# Physical Activity Levels and Sleep in Schoolchildren (6–17) with and without School Sport

**DOI:** 10.3390/ijerph20021263

**Published:** 2023-01-10

**Authors:** Arkaitz Larrinaga-Undabarrena, Xabier Río, Iker Sáez, Garazi Angulo-Garay, Aitor Martinez Aguirre-Betolaza, Neritzel Albisua, Gorka Martínez de Lahidalga Aguirre, José Ramón Sánchez Isla, Natalia García, Mikel Urbano, Myriam Guerra-Balic, Juan Ramón Fernández, Aitor Coca

**Affiliations:** 1Department of Physical Activity and Sport Science, Faculty of Education and Sport, University of Deusto, 48007 Bilbao, Spain; 2Department of Physical Activity and Health, Osasuna Mugimendua Kontrola S.L. Mugikon, 48450 Bilbao, Spain; 3Faculty of Humanities and Education Science, Mondragon University, 20500 Arrasate, Spain; 4Athlon Cooperative Society, 20500 Arrasate, Spain; 5Faculty of Health Sciences, University of Deusto, 48007 Bilbao, Spain; 6Faculty of Psychology, Education and Sport Sciences—Blanquerna, University Ramon Llull, 08022 Barcelona, Spain; 7Public College of Sports Teachings, Kirolene, Basque Government, 48200 Durango, Spain; 8Department of Physical Activity and Sports Sciences, Faculty of Health Sciences, Euneiz University, 01013 Vitoria-Gasteiz, Spain

**Keywords:** physical activity, sleep, schoolchildren, sedentary behaviour, school sport, accelerometer

## Abstract

There is strong evidence to support the association between daily physical activity and sleep parameters in children and adolescents. Physical activity and outdoor play are favourably associated with most sleep outcomes in school children. The aim is to find out the levels of physical activity and the quality of sleep in Basque schoolchildren aged between six and seventeen and to analyse the possible differences between those who carry out some kind of physical sports activity and those who do not. The sample consisted of 1082 schoolchildren (50.1% male and 49.9% female). Differences between groups were compared using the Mann–Whitney U test (2 samples) and Kruskal–Wallis one-factor ANOVA (k samples). A total of 723 (66.94%) of the participants said they practiced some physical sports activity. The accelerometers obtained significant differences in all levels of physical activity, as well as in sleep efficiency, with higher levels of physical activity (sedentary *p* = 0.001; light *p* = 0.017; moderate *p* = 0.009; vigorous *p* = 0.001 and moderate-to-vigorous physical activity *p* = 0.002) and better sleep efficiency (*p* = 0.002) in those schoolchildren who perform some type of physical sports activity. A significant difference in time spent in sedentary activities was also observed between primary and secondary school pupils of both sexes and regardless of the degree of physical sports activity completion.

## 1. Introduction

Low levels of physical activity are a serious health problem worldwide, being totally related to obesity and overweight [1,2]. The World Health Organization (WHO) states that physical inactivity is the fourth leading cause of death in the world [3], and physical inactivity is one of the ten leading causes of premature death, being directly responsible for many chronic diseases [4], which cause 38 million deaths annually [5].

Maintaining appropriate levels of physical activity in the first years of life seems to be fundamental for physical and mental development [6]. The WHO recommendation is that children and adolescents aged 5–17 years should engage in at least 60 min/day of moderate-to-vigorous physical activity, mainly aerobic [3]. It is also advised that it is important to practice vigorous physical activity, with a minimum of three days a week, since this strengthens muscles and bones. It is known that physical activity in sports has a positive impact on quality of life due to the psychological, physiological and social benefits it brings [7]. In fact, there is a direct relationship between the performance of physical activity, improved behaviour and its relationship with school performance [8,9], in addition to improving memory, attention span and/or executive functions [10]. Several studies demonstrate that regular physical activity has health benefits for everyone, regardless of body size, race, age, sex or ethnicity. Some benefits occur immediately, such as reduced feelings of anxiety, reduced blood pressure and improved sleep, cognitive function, and insulin sensitivity [11,12]. Other benefits, such as increased cardiorespiratory fitness, increased muscular strength, decreased depressive symptoms and sustained reduction in blood pressure, occur over months or years of being physically active [13].

In contrast, increased physical inactivity is associated with cardiovascular complications such as hypertension or metabolic diseases, which can lead to childhood obesity [9]. Physical inactivity at early ages may represent a major problem, in that childhood health behaviour patterns are likely to persist into adulthood [14]. Therefore, childhood and adolescence are key stages for adopting healthy lifestyles [15], as is the case of organised school sports practice. The implementation of a healthy lifestyle in which active play, free movement and organised sports activity are present can have a positive effect on child development and growth from an early age [16].

Worldwide, 80% of young people do not meet the minimum recommendations for physical activity, and in addition, many of them spend more than two hours a day in front of a screen [17]. In Spain, only three out of every ten boys and girls between eight and sixteen years of age fulfill the minimum WHO recommendations [18], which may be a consequence of the fact that the Spanish infant–juvenile population is the one that does the least physical-sporting activity outside school hours (not school hours) in comparison with other European countries [19]. One of the supporting data is that schoolchildren perform 24% more moderate or vigorous physical activity in educational interventions than those who do not [20]. Therefore, the importance of promoting physical activity in the school environment should be noted [21].

There is strong evidence supporting the associations between daily physical activity and sleep parameters [13,22], not only in the adult population but also in children and adolescents [23,24]. Moreover, reduced physical activity and increased screen time have shown to have an adversely impact older children’s sleep [25]. Indeed, that study showed that physical activity and outdoor play specifically were favourably associated with most sleep outcomes in toddlers and preschoolers.

There is an association between sleep duration and the risk of being overweight or obese [26,27] in both children [28,29] and adolescents [7,30]. Adolescence is marked by dramatic changes in sleep [31]; there is in fact a high prevalence of insufficient and poor-quality sleep [32]. In addition, the hours of sleep have been shown to be of vital importance in different physiological and psychological processes [33]. Deprivation of these sleep hours affects the level of arousal during waking hours, hindering active and efficient behaviour [34]. Waking and sleeping cycles change markedly during childhood due to hormonal influences on melatonin secretion and the regulatory mechanisms of the sleep–wake cycle [35]. For this reason, physical activity is important because it increases melatonin levels and can have a positive impact on the reduction of insomnia [36]. Establishing an active routine and reducing light/blue light rays from electronic devices in the hours before sleep are associated with better sleep reconciliation, fewer nighttime awakenings, and increased sleep duration [37]. Moreover, the more time spent with electronic devices, the lower the levels of physical activity [38], thus increasing levels of sedentary behaviour. Moreover, the evolution of technology has been accelerated during these last two years as a result of COVID-19 [39], causing great changes in the behaviours of the population and having a direct impact both at the emotional and psychological level [40], as well as on the physical activity levels of children and adolescents [14,41]. In fact, the percentage of inactive people increased considerably [42,43], making the situation even worse.

Several European studies [44,45] that measure physical activity in children and adolescents detect that in most cases the minimum recommended levels of physical activity are not met, showing the need to implement a common framework for systematic monitoring of physical activity indicators in children and young people, which is necessary to begin to control the change over time and the efficiency of fieldwork.

Thus, the main objective of this study is to know the levels of physical activity and the quality of sleep in schoolchildren between six and seventeen years of age in the Autonomous Community of the Basque Country and to analyse the possible differences between those who perform some type of formal physical activity and those who do not perform formal physical activity.

## 2. Materials and Methods

### 2.1. Subjects and Design

A cross-sectional observational study was carried out on a sample of 1082 schoolchildren between 6 and 17 years of age in the Basque Country (50.1% male and 49.9% female). A proportional and random stratification according to province and region, sex and age was taken into account. The participating schools were randomly selected according to the aforementioned criteria.

### 2.2. Instruments and Variables

The ActiGraph WGT3X-BT accelerometer (manufacturer ActiGraph, 49 East Chase, St. Pensacola, FL, USA) was used to collect data related to physical activity levels, including moderate and vigorous physical activity, as well as sleep parameters. Participants wore the ActiGraph WGT3X-BT for seven consecutive days, including a weekend. The device was worn on the wrist of the nondominant hand, considering valid recordings with a minimum of 10 h per day, at least 3 days of which were 2 school days and one weekend day. The participants were asked to dispense with the device during bathing, showering and/or water activities. The record of physical activity (time and type), sleep time and its efficiency [46,47,48] were collected in the most valid and reliable way [49]. In addition, like other authors, a diary was filled out that collected data such as sleep time, mode of transportation to school or other places and the type of physical activity they perform [50]. Waking hours were classified as sedentary time, light (LPA), moderate (MPA), vigorous (VPA), and moderate-to-vigorous physical activity (MVPA) [51] recommended as cut-points from previous research [52].

The variables under study were sex (male/female), academic year (Primary Education from 1st to 6th grade, Secondary Education from 1st to 4th grade and High School from 1st to 2nd grade) and type of school (public and private), as population variables; participation and nonparticipation in physical sports activity (PSA) as independent variables; and finally, physical activity levels (LPA, MPA, VPA and MVPA), sedentary behaviour (min), total time in bed (min), total sleep time (min), nighttime awakenings (min) and sleep efficiency (%) as dependent variables.

### 2.3. Procedure

To conduct the research, approval was requested from the Basque Medicines Research Ethics Committee (Department of Health of the Basque Government) in accordance with Law 14/2007 on biomedical research [53]. In the same way, the Ethical principles of the Declaration of Helsinki [54] and other applicable ethical and legislative principles were respected from the report of the Basque Medicines Research Ethics Committee; approval was obtained with the internal code PI2020011.

In the same way, the current regulations on personal data protection were respected: Regulation (EU) 2016/679 of 27 April 2016 (GDPR) [55], Organic Law 3/2018 of 5 December on Personal Data Protection and guarantee of digital rights [56], and Royal Decree 1720/2007 of 21 December [57].

Once the approval of the project was received, the Department of Education of the Basque Government launched a mass email to all schools in the Basque Autonomous Community. In case of a positive response, interviews were arranged with the centre’s management team, from which a person responsible for the student selection process was selected, as well as for providing information to the families. Once the students were selected, a draw was made among the participants who met the selection criteria, whose parents signed their consent and the students themselves their informed assent. Finally, the days on which the accelerometers were to be placed were established, together with their subsequent removal.

There was a proportional and random stratification according to province and county, age, sex, educational network (public or private), together with inclusion and exclusion criteria, which can be seen in Table 1.

### 2.4. Statistical Analysis

Statistical analysis was performed with SPSS software (version 28.0.1.0; IBM Corp., New York, NY, USA). Values of *p* < 0.05 were considered statistically significant. First, the Kolmogorov–Smirnov test was used to evaluate the normality of the distribution and the Levene contrast to observe the homogeneity of variances, as well as an analysis of the descriptive variables studied (means, standard deviation, etc.). None of the variables studied met the above requirements, so differences between groups were compared using the nonparametric Mann–Whitney U test (2 samples) and Kruskal–Wallis one-factor ANOVA (k samples).

For outcome variables, descriptive statistics were used, reporting the level of significance corresponding to the main group (between participants). To avoid a type I error, a post hoc analysis was performed when the interaction effect was significant. Values are expressed as mean (SD).

## 3. Results

The total sample reported was the entire school population of 111 public and private schools in the Basque Country, of which 1082 schoolchildren (542 boys and 540 girls) were recruited in the 75 schools that gave their definitive approval to the study. Of the total sample recruited, 723 (66.94%) said they participated in some PSA, while 359 (33.05%) said they did not participate in any PSA. As can see in Table 2, the sleep efficiency, the total bed time, the total sleep time and WASO between organised and non-organised sports.

The results obtained through the accelerometers obtained significant differences in all levels of physical activity (Table 3), as well as in sleep efficiency (Table 4), with higher levels of physical activity (sedentary *p* = 0.001; light *p* = 0.017; moderate *p* = 0.009; vigorous *p* = 0.001; and MVPA *p* = 0.002) and better sleep efficiency (*p* = 0.002) in those schoolchildren who perform some type of sports physical activity than those who do not perform regulated sports physical activity.

The descriptive results of the sleep variables of the entire sample segregated by academic year and sex can be examined in Table 4. Regarding total time in bed and sedentary behaviour, those who do not perform PSA show a greater time in bed (480.41 min ± 79.53 vs. 491.58 min ± 88.12; *p* = 0.04) as well as higher values of sedentary behaviour (556.67 min (179.81) vs. 620.56 min (178.39)), these being significant with respect to those who perform some regulated PSA (*p* = 0.001). A significant difference was also observed in the time spent in sedentary activities between primary and secondary school students for both sexes and independently of the performance of PSA (Table 5).

According to the age criterion, all sleep parameters are higher in primary school children (6–12 years) than in adolescents in secondary and high school (12–17 years), irrespective of sex. However, sleep efficiency is higher in adolescents. In addition, differences have been observed in primary school children compared to the adolescent group in all physical activity ranges for both males and females. The primary school children have less sedentary behavior and perform more physical activity than the adolescent group in all ranges.

With regard to school ownership, although this was not one of the variables under study, some significant effects on sleep and levels of physical activity were detected in schoolchildren (Table 6 and Table 7). In addition to what is described in Table 6 and Table 7, the total sleep time seems to be higher in public school students than in private schools, both for men (public 419.07 min ± 54.99 vs. private 413.58 min ± 67.84; *p* = 0.017) and women (public 427.05 min ± 69.18 vs. private 407.22 min ± 65.82; *p* = 0.011). In the latter, the total time in bed is also greater when they are attending a public school (public 490.33 min ± 82.22 vs. private 469.89 min ± 80.42; *p* = 0.024). Differences were also observed in males who do not perform PSA compared to total sleep time (public 435.41 min ± 91.85 vs. private 408.22 min ± 64.47; *p* = 0.028), but not in females. Among male PSA students, school ownership seems to affect sleep efficiency (public 85.92% ± 6.03 vs. private 88.03% ± 5.73; *p* < 0.001) and WASO (public 63.22 min ± 32.77 vs. private 57.06 ± 32.87; *p* = 0.047), being better in public schools.

There are also differences between men in public and private schools in time spent in sedentary activities (public 539.34 min ± 183.09 vs. private 585.38 ± 179.43; *p* = 0.028) for the group of those doing PSA, and in light physical activity exclusively for those not performing PSA (public 195.33 min ± 61.11 vs. private 208.35 ± 62.77; *p* = 0.035).

## 4. Discussion

Physical activity is any bodily movement produced by skeletal muscles that requires energy expenditure [58]. Levels of physical activity were analysed in the present study among those who practice some type of PSA, this term encompassing all structured, institutionalised motor activities presented in a school environment, not within the subject of physical education, in the form of competition or concentration/exhibition [59,60]. These activities can be included within school educational initiatives and in nonschool hours, where motor play prevails over adult sports and their competitions. There is extensive literature on the terminological conception of motor action, its classification and derivations to sporting activities [61,62,63,64]. In the present study, we focused on collecting the levels of physical activity quantitatively and asking which of the participants were involved in some type of school sports activity (regulated, structured, recreational and/or competitive), introducing those traditional games organised within this term, although there is a difference between them [65,66,67].

The ActiGraph WGT3X-BT sensors, used in this study to quantitatively measure sleep values and physical activity levels at school age, have been widely used in research and have demonstrated reproducibility, validity and feasibility for children and adolescents [13,68,69].

Using a representative sample of children and adolescents in the Basque Country (at 95% confidence level and with a margin of error of 5%), like other authors [45], a positive impact was observed of physical activity on physical activity variables, promoting normal growth and development and helping people to feel, function and sleep better, as well as reducing the risk of many chronic diseases.

The results obtained in the present study, in terms of participation in physical activity, are similar to those obtained in other European studies (66.7% vs. 66%) [70]. There are differences in the age group 6–12 years compared to 12–17 years in all physical activity ranges in both males and females. Children aged 6–12 years have less sedentary behaviour and perform more physical activity than those aged 12–17 years in all ranges, which may be related to the competitive fact that as the cognitive ability to assess one’s own ability in relation to the abilities of others develops, it influences perceived competence and motivation to participate and its psychological consequences [71]. In the present study, it was observed that children and adolescents in the Basque Country do not comply with the recommendations of the American Academy of Sleep Medicine (±420 min). As can be seen in Table 3 and Table 4, all sleep parameters are higher in children aged 6–12 years than in adolescents aged 12–17 years, as in other studies [72,73], irrespective of sex. This may be related to increased pressure, stress and anxiety in adolescence due to academic and other reasons [74,75]. However, sleep efficiency is higher in adolescents (12–17 years) in contrast to other studies where sleep efficiency parameters decline with age [72,76] which would not represent a sleep disorder problem in this group (12–17 years) [77]. These results could be a macrolevel indicator of differences in social environments (socioeconomic status) and social demands (school, as well as cultural attitudes and values regarding sleep and its importance) [72]. Higher physical activity MVPA values in the 6–12-year group relative to the 12–17-year group may be related to poorer outcomes in recorded sleep parameters, and a multinational study of 5779 children aged 9–11 years found that moderate-to-vigorous-intensity physical activity measured with an actygraph at the waist was associated with longer sleep duration measured with the same device [78].

As can be seen in Table 2, our results suggest that spending more time in bed (480.4 vs. 491.6 min) and/or sleeping more (416.6 vs. 420.9 min) does not translate into better sleep quality and efficiency (87.2 vs. 86.1). Performing more physical activity seems to be determinant in this aspect, a fact that has already been demonstrated by several authors [74,79,80].

The results of this study are in line with others that show that moderate or vigorous physical activity improves sleep efficiency in school-age children, with a bidirectional association, whereby increasing moderate or vigorous physical activity increases sleep duration [77,81,82] and efficiency [77]. Similarly, in a study of adolescents, it was observed that those who ran every morning for 30 min for 3 consecutive weeks showed improvements in objective measures of sleep, along with a decrease in sleep latency [83]. This may correspond to the results of the present study, with significantly greater sleep efficiency in children and adolescents who practice PSA, compared to those who do not. Other research also affirms that a longer duration of weekly exercise is associated with a longer sleep onset latency and fewer deep awakenings [84]. A solution to improve the quality of sleep in children and adolescents can be found in the promotion of activities such as encouraging outdoor play or organised sports leagues [85], coinciding with the results of the present study.

As in other studies, children and adolescents who engage in physical activity provide a high percentage of the total moderate and vigorous physical activity [86,87,88,89]. Taking into account the volume and the longer duration than formal physical education and recreation, out-of-school physical activity seems to be the main contributor [90].

Globally, there appear to be marked socioeconomic disparities between students in public and public schools [91]. Any walking and any MVPA did not differ between school types for high school or middle school parents [92]; however, the results of the present study show data in favour of public school ownership in total sleep time and time in bed as well as lower sedentary behaviour, but not in sleep efficiency in those performing PAS, relative to privately owned schools. There were also differences in the group that did not perform PSA in LPA in favour of public school. This may be due to the proximity and geographical location of public schools in the Basque Country, making it easier to walk to all of them. In private schools, there are a high percentage of schools on the outskirts of population centres or cities, which makes it difficult for them to engage in daily physical activity on the way to the school, and this could be a determining factor in the differential values of APL in those who do not do PSA. On the other hand, sleep values may be related to the pressure, anxiety and amount of homework [93] that the private system, as opposed to the public system, imposes on its students, which, as the literature indicates, increases the probability of suffering from sleep disorders, along with sedentary behaviour [91].

The limitations of the present study are limited, on the one hand, to the collection of qualitative information when completing the personal diary that was requested from each participant; although it is detailed whether they participate in any PSA, it is not known in which activities and/or sports they practice. In the present study, it is requested that the accelerometer be removed from the wrist in the case of water activities and baths or showers, so there is very interesting information that is not collected in practitioners of water sports, whether regulated or for leisure.

The present study suggests this field as an important future line of research, since there is a lack of studies estimating moderate and vigorous physical activity that report these data in children between 5 and 11 years of age. The increase in sedentary behaviours in children and adolescents, especially in the case of women, demands the proposal of lines of intervention consistent with the current social reality that guide the development of projects to promote physical activity. For this purpose, despite the fact that at the curricular level in different subjects (especially in physical education), contents related to healthy lifestyle, physical condition and the life hygiene of schoolchildren are addressed, it seems not to be enough to improve life habits, so it is necessary to develop new interdisciplinary or cross-cutting curricular proposals.

## 5. Conclusions

Subjects who performed PSA were found to have significant positive differences in both sleep efficiency, time in bed and dedicatory behaviour compared to those who did not perform PSA. However, no differences were observed in total time asleep or wakefulness after sleep onset. A significant difference in time spent in sedentary activities was also observed between primary and secondary school pupils of both sexes and regardless of the degree of PSA completion, with pupils aged 6 to 12 years having less sedentary behaviour and performing more physical activity than those aged 12 to 17 years in all ranges.

The total sleep time of Basque schoolchildren is higher in public schools than in private schools, both for boys and girls. In the latter, the total time in bed is also longer when they attend a public school.

Not performing PAS with respect to total sleep time has significant differences between private and public schools for boys, but not for girls. Among male students who perform PAS, school ownership affects sleep efficiency and WASO, with better values obtained in public schools than in private schools. Boys in public schools have lower sedentary behaviour for the PAS group, and in light physical activity exclusively for non-PAS students.

The promotion of school sport represents an opportunity for public authorities to improve the health of the community.

## Figures and Tables

**Table 1 ijerph-20-01263-t001:** Inclusion and exclusion criteria.

Inclusion criteria	Belong to the student body of a participating school or institute.
	Have authorisation to participate through a signed informed consent by the parents or legal guardians of the child or adolescent.
Exclusion criteria	Non-consent or refusal by the child or adolescent to complete the PA diary or to use the accelerometer, even with a signed informed consent by their parents or legal guardians.
	Physical or intellectual disability that prevents completing the daily PA or use of the accelerometer according to the defined protocol. Each case will be assessed with each school’s teaching team and with the parents or legal guardians of the minor.

**Table 2 ijerph-20-01263-t002:** Results of the daily sleep quality analysis.

	Practice Organised/Federated Sports (*n* = 723)	Do Not Practice Organised/Federated Sports (*n* = 359)	*p* between Groups Title 3
Sleep efficiency (%)	87.16 ± 5.84	86.12 ± 6.81	0.022
Total bed time (min)	480.41 ± 79.53	491.58 ± 88.12	0.040
Total sleep time (min)	416.59 ± 65.00	420.93 ± 74.59	0.455
WASO (min)	59.75 ± 31.40	64.36 ± 35.94	0.056

Data are presented as mean or percentage ± standard deviation (SD). WASO: wake after sleep onset.

**Table 3 ijerph-20-01263-t003:** Results of the daily physical activity (PA) levels and sedentary behaviour.

	Practice Organised/Federated Sports (*n* = 723)	Do Not Practice Organised/Federated Sports (*n* = 359)	*p* between Groups Title 3
Sedentary (min)	556.67 ± 179.81	620.56 ± 178.39	0.001
Light PA (min)	215.29 ± 66.34	209.92 ± 59.21	0.017
Moderate PA (min)	81.05 ± 37.47	75.62 ± 35.07	0.009
Vigorous PA (min)	9.28 ± 7.97	7.13 ± 6.72	0.001
MVPA (min)	90.33 ± 43.47	82.75 ± 40.57	0.002

Data are presented as mean or percentage ± standard deviation (SD). MVPA: moderate to vigorous physical activity.

**Table 4 ijerph-20-01263-t004:** Results of daily sleep quality analysis in each of the school years between practitioners and nonpractitioners of organised/federated sports in each sex.

				BMI (kg/m^2^)		Sleep Efficiency (%)		Total Bed Time (min)		Total Sleep Time (min)		WASO (min)	
Male	Fed. Sport	Yes	No	Yes	No	Yes	No	Yes	No	Yes	No	Yes	No
Primary Education	1st (*n* = 55)	39	16	17.10 ± 2.30	16.58 ± 2.63	85.32 ± 6.14	81.65 ± 6.13	489.10 ± 97.41	548.45 ± 89.67	412.29 ± 65.57	443.17 ± 56.19	69.76 ± 40.11	89.04 ± 41.67
	2nd (*n* = 42)	27	15	16.48 ± 2.03	17.71 ± 3.07	86.26 ± 6.04	82.89 ± 6.54	515.58± 83.94	533.69 ± 68.00	443.65 ± 74.04	439.81 ± 45.13	62.99± 27.90	84.70 ± 39.74
	3rd (*n* = 55)	37	18	17.44 ± 3.99	17.02 ± 3.10	86.18 ± 5.94	83.36 ± 7.66	508.25 ± 72.51	529.38 ± 95.59	435.38 ± 52.09	435.89 ± 61.65	67.17 ± 33.19	86.64 ± 47.73
	4th (*n* = 54)	37	17	17.72 ± 3.14	17.87 ± 2.59	86.41 ± 7.09	86.27 ± 5.06	506.20± 83.99	503.52 ± 107.14	436.46 ± 75.85	435.74 ± 99.52	63.30 ± 37.44	59.09 ± 28.58
	5th (*n* = 42)	34	8	17.97 ± 2.57	18.03 ± 3.35	88.26 ± 4.58	86.74 ± 7.02	507.95 ± 60.89	503.69 ± 133.32	445.67 ± 40.41	436.15 ± 130.42	64.18 ± 35.13	58.83 ± 32.81
	6th (*n* = 49)	34	15	18.77 ± 3.29	19.40 ± 2.15	85.48 ± 6.24	87.48 ± 7.93	479.36 ± 81.17	487.51 ± 98.57	408.62 ± 60.97	425.33 ± 96.15	63.55 ± 33.67	62.46 ± 47.55
Secondary Education	1st (*n* = 39)	27	12	19.66 ± 3.54	18.74 ± 2.19	86.50 ± 7.16	86.43 ± 7.80	476.30 ± 58.40	475.61 ± 68.79	409.87 ± 46.93	408.28 ± 48.93	61.93 ± 37.25	61.84 ± 40.36
	2nd (*n* = 42)	34	8	23.42 ± 2.86	19.80 ± 2.59	87.39 ± 6.46	85.29 ± 3.84	472.12 ± 58.40	444.98 ± 100.89	412.32 ± 76.97	378.27 ± 86.87	54.83 ± 26.71	68.09 ± 18.20
	3rd (*n* = 40)	24	16	20.67 ± 3.52	19.50 ± 2.30	87.97 ± 4.41	86.64 ± 5.69	446.70 ± 53.73	446.27 ± 69.54	390.65 ± 39.81	386.52 ± 61.05	51.07 ± 24.85	56.96 ± 26.49
	4th (*n* = 50)	30	20	20.46 ± 2.89	21.34 ± 3.17	90.09 ± 4.03	86.64 ± 7.92	434.68 ± 64.72	485.84 ± 75.02 ^#^	391.36 ± 58.69	425.78 ± 95.72	44.42 ± 22.83	56.03 ± 35.95
High School	1st (*n* = 38)	25	13	21.65 ± 2.19	20.23 ± 2.17	87.34 ± 6.58	86.72 ± 8.06	463.75 ± 71.83	473.40 ± 93.04	402.09 ± 50.98	409.12 ± 95.07	56.49 ± 35.44	54.74 ± 29.32
	2nd (*n* = 36)	18	18	21.37 ± 3.69	20.79 ± 1.79 ^#^	87.99 ± 4.31	87.73 ± 10.17	432.64 ± 37.12	483.96 ± 77.0	379.25 ± 30.29	423.36 ± 85.35	48.86 ± 21.64	54.12 ± 45.20
Total	6–12 yrs. old 12–17 yrs. old	208 158	89 87	17.60 ± 3.00 * 21.20 ± 1.00 *	17.70 ± 2.80 * 20.20 ± 2.50	86.30 ± 6.07 * 87.90 ± 5.75	84.53 ± 6.92 * ^#^ 86.72 ± 7.64	500.47 ± 81.01 * 456.04 ± 64.17	519.23 ± 96.80 * ^#^ 471.14 ± 78.53	429.62 ± 63.36 * 399.24 ± 55.69	436.07 ± 79.07 * 408.79 ± 81.16	65.34 ± 34.82 * 53.08 ± 29.02	74.91 ± 41.95 * 57.52 ± 34.42
Primary Education	1st (*n* = 51)	26	25	16.19 ± 81.83	15.99 ± 2.24	86.55 ± 5.99	84.63 ± 6.07	529.23 ± 95.24	510.06 ± 100.97	454.95 ± 75.97	428.73 ± 83.25	70.10 ± 34.02	72.41 ± 34.74
	2nd (*n* = 54)	36	18	16.73 ± 2.28	16.75 ± 1.55	86.58 ± 5.59	83.20 ± 6.44 ^#^	491.93 ± 97.66	507.57 ± 102.47	420.90 ± 69.42	417.31 ± 66.11	67.38 ± 33.70	78.31 ± 39.53
	3rd (*n* = 51)	32	19	17.07 ± 2.21	17.07 ± 2.27	86.42 ± 4.37	85.17 ± 5.98	504.36 ± 84.73	541.15 ± 77.13	435.13 ± 75.39	458.08 ± 57.39	60.86 ± 24.67	77.14 ± 30.74
	4th (*n* = 52)	35	17	18.98 ± 2.86	18.46 ± 3.28	68.71 ± 5.73	87.65 ± 6.74	480.60 ± 104.10	511.47 ± 83.37	414.73 ± 87.61	44.99 ± 69.46	66.70 ± 33.40	64.81 ± 32.72
	5th (*n* = 42)	32	10	18.54 ± 3.10	17.35 ± 2.07	87.48 ± 4.14	88.35 ± 4.49	484.39 ± 76.20	480.64 ± 92.80	422.02 ± 57.71	422.04± 79.99	63.15 ± 24.68	54.21 ± 21.44
	6th (*n* = 46)	32	14	17.85 ± 2.35	18.17 ± 2.58	86.17 ± 6.14	88.02 ± 4.33	478.27 ± 94.82	452.49 ± 84.70	409.14 ± 72.72	397.10 ± 76.26	62.48 ± 30.88	54.81 ± 21.05
Secondary Education	1st (*n* = 40)	23	17	19.56 ± 2.10	18.52 ± 2.49	89.74 ± 5.12	84.69 ± 8.02	490.51 ± 82.28	478.27 ± 83.50	439.02 ± 74.71	400.72 ± 58.69	49.08 ± 25.60	71.09 ± 41.88
	2nd (*n* = 41)	27	14	20.46 ± 4.56	20.35 ± 2.20	85.95 ± 7.65	88.47 ± 5.32	475.37 ± 57.91	467.88 ± 69.49	406.48 ± 50.23	413.80 ± 68.80	62.72 ± 34.71	51.57 ± 25.55
	3rd (*n* = 42)	32	10	20.48 ± 2.33	19.11 ± 2.22	87.87± 5.75	87.82 ± 6.50	468.00 ± 54.52	458.05 ± 59.30	410.24 ± 59.22	402.18 ± 61.10	54.96 ± 25.27	53.83 ± 29.43
	4th (*n* = 45)	32	13	20.56 ± 2.93	22.40 ± 3.46	87.55 ± 5.12	87.64 ± 5.82	426.98 ± 59.92	468.40 ± 60.91 ^#^	372.28 ± 49.24	408.25 ± 47.70 ^#^	51.82 ± 23.45	54.53 ± 26.14
High School	1st (*n* = 43)	28	15	21.04 ± 2.95	21.44 ± 2.35	90.29 ± 5.52	90.64 ± 5.64	464.68 ± 58.44	467.26 ± 62.98	418.71 ± 59.55	424.05 ± 66.47	43.42 ± 27.88	39.50 ± 23.09
	2nd (*n* = 33)	23	10	21.26 ± 2.33	21.71 ± 3.80	87.44 ± 6.56	86.57 ± 5.36	469.52 ± 52.58	442.91 ± 75.56	408.69 ± 44.21	380.01 ± 52.31	54.91 ± 30.25	51.73 ± 23.55
Total	6–12 yrs old	193	103	17.60 ± 2.60 *	17.10 ± 2.40	86.65 ± 5.32 *	85.80 ± 6.06	493.45 ± 93.08 *	504.91 ± 93.05 *	424.96 ± 74.18 *	429.22 ± 72.78 *	65.03 ± 30.27 *	68.90 ± 32.53 *
	12–17 yrs old	165	79	20.50 ± 2.90	20.50 ± 3.00	88.10 ± 6.07	87.61 ± 6.40	464.04 ± 63.31	465.68 ± 68.51	407.49 ± 59.33	406.27 ± 59.62	52.92 ± 28.08	54.27± 30.65

Data are presented as mean ± SD. Primary Education: from 6 to 12 years old; Secondary Education: from 12 to 15 years old; High School: from 15 to 17 years old; Yes: do practice organised/federated sports; No: do not practice organised/federated sports; PA: physical activity; ^#^ Statistical intragroup differences between practitioners and nonpractitioners of organised/federated sports. * Statistical intergroup differences between 6–12- and 12-to-17-year-old groups in practitioners and nonpractitioners of organised/federated sports. * *p* ≤ 0.05.

**Table 5 ijerph-20-01263-t005:** Results of the daily physical activity levels and sedentary behavior in each of the school years between practitioners and nonpractitioners of organised/federated sports in each sex.

				Sedentary (min)		Light PA (min)		Moderate PA (min)		Vigorous PA (min)		MVPA (min)	
Male	Fed. Sport	Yes	No	Yes	No	Yes	No	Yes	No	Yes	No	Yes	No
Primary Education	1st (*n* = 55)	39	16	487.17 ± 142.14	542.79 ± 130.59	251.17 ± 56.75	259.20 ± 29.33	100.15 ± 38.98	120.52 ± 21.17	15.26 ± 9.23	17.02 ± 6.03	115.41 ± 46.50	137.54 ± 26.01
	2nd (*n* = 42)	27	15	486.72 ± 151.33	535.50 ± 201.02	248.65 ± 57.71	230.75 ± 69.11	110.42 ± 29.79	102.74 ± 42.82	17.07 ± 8.08	16.15 ± 11.7	127.50 ± 35.45	118.90 ± 53.27
	3rd (*n* = 55)	37	18	528.60 ± 112.85	628.59 ± 197.68	233.35 ± 50.58	216.45 ± 64.65	97.30 ± 28.26	86.08 ± 30.96	15.15 ± 7.83	11.88 ± 7.21	112.45 ± 34.64	97.96 ± 37.07
	4th (*n* = 54)	37	17	552.27 ± 146.38	557.46 ± 102.40	231.41 ± 44.78	234.25 ± 32.99	102.40± 27.88	93.03 ± 25.19	15.86 ± 6.91	11.95 ± 5.66 ^#^	118.27 ± 32.09	104.99 ± 29.89
	5th (*n* = 42)	34	8	564.79 ± 101.33	688.74 ± 148.97 ^##^	238.68 ± 37.44	199.62 ± 38.43	97.10 ± 24.56	76.39 ± 24.94	14.68 ± 9.95	10.70 ± 7.58	111.78 ± 30.93	87.09 ± 32.16
	6th (*n* = 49)	34	15	574.70 ± 168.74	611.51 ± 139.11	215.99 ± 60.07	201.56 ± 35.40	84.29 ± 34.12	70.03 ± 28.10	11.94 ± 8.70	9.47 ± 8.11	96.24 ± 41.32	79.49 ± 35.56
Secondary Education	1st (*n* = 39)	27	12	524.97 ± 222.0	608.74 ± 222.47	198.19 ± 81.94	192.55 ± 68.34	68.65 ± 35.81	61.50 ± 27.51	8.42 ± 8.86	5.15 ± 4.37	77.07 ± 41.92	66.65 ± 30.56
	2nd (*n* = 42)	34	8	628.36 ± 178.53	746.23 ± 124.58	200.72 ± 56.45	214.74 ± 27.09	62.19 ± 27.50	63.83 ± 14.67	6.12 ± 6.13	4.43 ± 3.14	68.31 ± 31.94	68.27 ± 16.16
	3rd (*n* = 40)	24	16	681.38 ± 166.29	660.73 ± 166.24	200.08 ± 35.14	191.74 ± 65.92	60.17 ± 33.21	55.84 ± 27.00	7.32 ± 7.13	3.43 ± 3.98 ^#^	67.50 ± 39.88	59.28 ± 29.59
	4th (*n* = 50)	30	20	621.05 ± 187.93	647.09 ± 218.47	166.35 ± 59.75	164.53 ± 64.94	42.54 ± 21.72	41.42 ± 24.41	4.80 ± 7.64	3.33 ± 3.61	47.35 ± 26.58	44.75 ± 26.40
High School	1st (*n* = 38)	25	13	594.54 ± 252.66	631.65 ± 308.41	166.42 ± 71.97	137.29 ± 69.42	46.00 ± 28.11	28.22 ± 14.86 ^#^	5.25 ± 4.86	5.79 ± 6.09 ^##^	51.25 ± 32.43	28.80 ± 15.29 ^#^
	2nd (*n* = 36)	18	18	575.98 ± 288.67	635.67 ± 161.34	160.26 ± 82.09	176.63 ± 44.71	38.84 ± 23.98	40.15 ± 23.91	2.37 ± 2.82	2.34 ± 5.02	41.21 ± 25.76	42.50 ± 28.40
	6–12 yrs old	208	89	533.06 ± 141.06 *	586.42 ± 161.27 * ^##^	236.36 ± 52.33 *	225.92 ± 51.28 *	98.29 ± 31.65 *	92.8 ± 33.5 *	14.94 ± 8.54 *	13.02 ± 8.17 * ^##^	113.23 ± 38.04 *	105.86 ± 40.66 * ^#^
Total	12–17 yrs old	158	87	606.04 ± 215.62	648.76 ± 206.38 ^#^	183.63 ± 66.72	176.45 ± 62.41	54.03 ± 30.36	46.67 ± 25.64 ^#^	5.88 ± 6.84	3.08 ± 3.82 ^###^	59.92 ± 35.47	49.76 ± 28.48 ^#^
Primary Education	1st (*n* = 51)	26	25	427.02 ± 170.58	527.73 ± 208.19 ^##^	248.49 ± 80.50	241.11 ± 78.58	103.16 ± 46.31	99.85 ± 38.38	12.19 ± 7.90	10.16 ± 5.25	115.36 ± 52.20	110.02 ± 42.89
	2nd (*n* = 54)	36	18	439.28 ± 125.80	602.54 ± 101.09 ^###^	244.26 ± 70.3	261.39 ± 26.60	107.05 ± 41.57	104.41 ± 19.65	11.05 ± 7.80	9.03 ± 3.72	118.11 ± 47.99	113.44 ± 21.45
	3rd (*n* = 51)	32	19	496.97 ± 131.48	585.24 ± 118.40 ^#^	244.72 ± 59.69	245.85 ± 24.81	89.79 ± 32.18	105.37 ± 19.10	10.95 ± 6.66	11.85 ± 4.05	109.74 ± 37.02	117.22 ± 22.04
	4th (*n* = 52)	35	17	463.82 ± 153.52	598.28 ± 146.27 ^#^	233.80 ± 76.26	243.30 ± 38.31	95.55 ± 34.08	95.80 ± 26.45	8.87 ± 4.76	7.96 ± 3.62	104.42 ± 37.39	103.76 ± 28.39
	5th (*n* = 42)	32	10	582.29 ± 122.36	624.32 ± 106.91	235.98 ± 40.92	216.18 ± 19.30	89.79 ± 28.17	82.37 ± 12.89	8.22 ± 4.59	6.02 ± 3.01	98.01 ± 29.37	88.39 ± 15.47
	6th (*n* = 46)	32	14	580.91 ± 158.36	607.71 ± 69.91	216.18 ± 65.73	229.22 ± 27.95	85.40 ± 33.64	83.91 ± 17.79	7.93 ± 6.15	5.39 ± 2.97	93.33 ± 36.92	89.30 ± 19.77
Secondary Education	1st (*n* = 40)	23	17	575.14 ± 210.64	648.26 ± 229.44	195.16 ± 70.56	187.42 ± 59.12	65.90 ± 30.27	64.90 ± 29.83	4.40 ± 3.96	3.90 ± 3.44	70.30 ± 33.18	68.81 ± 32.52
	2nd (*n* = 41)	27	14	574.66 ± 188.46	696.97 ± 150.85 ^##^	202.25 ± 68.30	197.81 ± 37.99	71.16 ± 28.62	71.65 ± 20.28	5.13 ± 4.80	2.91 ± 2.50	76.30 ± 31.92	74.57 ± 21.80
	3rd (*n* = 42)	32	10	617.26 ± 200.92	722.54 ± 123.51	195.09 ± 65.03	189.32 ± 44.10	69.01 ± 33.97	53.12 ± 18.59	5.11 ± 4.83	2.25 ± 2.85 ^#^	74.13 ± 37.74	55.37 ± 19.50
	4th (*n* = 45)	32	13	638.86 ± 156.06	739.51 ± 127.26	191.24 ± 53.97	188.51 ± 23.08	68.77 ± 29.19	60.61 ± 20.27	5.54 ± 5.59	3.55 ± 3.18	74.32 ± 32.40	64.17 ± 22.01
High School	1st (*n* = 43)	28	15	632.17 ± 130.83	643.80 ± 199.15	201.30 ± 40.89	180.25 ± 57.37	68.81 ± 24.89	55.69 ± 24.88	4.89 ± 3.42	2.06 ± 1.66 ^##^	73.70 ± 26.50	57.75 ± 25.54
	2nd (*n* = 33)	23	10	632.87 ± 161.75	680.83 ± 149.67	190.46 ± 60.12	198.96 ± 4.96	62.77 ± 34.34	56.06 ± 20.09	3.96 ± 4.80	1.25 ± 1.40^#^	66.73 ± 38.29	57.31 ± 21.13
Total	6–12 yrs. old	193	103	498.84 ± 156.37 *	583.31 ± 143.07 * ^###^	236.98 ± 66.56 *	241.85 ± 46.90 *	96.62 ± 36.40 *	97.13 ± 26.64 *	9.80 ± 6.51 *	8.86 ± 4.51 *	106.43 ± 41.01 *	106.00 ± 29.99 *
	12–17 yrs.old	165	79	613.59 ± 175.64	684.41 ± 172.82 ^##^	195.93 ± 59.48	189.78 ± 45.71 ^#^	67.98 ± 29.98	61.03 ± 23.49 ^#^	4.90 ± 4.70	2.77 ± 2.74 ^###^	72.88 ± 33.14	63.81 ± 25.12 ^##^

Data are presented as mean ± SD. Primary Education: from 6 to 12 years old; Secondary Education: from 12 to 15 years old; High School: from 15 to 17 years old; Yes: do practice organised/federated sports; No: do not practice organised/federated sports; PA: physical activity; ^#^ Statistical intragroup differences between practitioners and nonpractitioners of organised/federated sports. ^#^ *p* ≤ 0.05; ^##^ *p* ≤ 0.01; ^###^ *p* ≤ 0.001. * Statistical intergroup differences between 6–12- and 12-to-17-year-old groups in practitioners and nonpractitioners of organised/federated sports. * *p* ≤ 0.05.

**Table 6 ijerph-20-01263-t006:** Results of sleep quality variables on the basis of centre ownership and sex as independent variables of analysis.

				Sleep Efficiency (%)		Total Bed Time (min)		Total Sleep Time (min)		WASO (min)	
	Type School	Pu.	Pri.	Pu.	Pri.	Pu.	Pri.	Pu.	Pri.	Pu.	Pri.
Male	(*n* = 542)	271	271	86.06 ± 6.57	87.28 ± 6.18 *	493.95 ± 83.80 *	472.26 ± 80.43	423.40 ± 70.89 *	410.02 ± 66.77	62.58 ± 34.32	60.36 ± 35.29
Female	(*n* = 540)	275	265	87.29 ± 6.39 ^#^	86.93 ± 5.58	486.53 ± 86.43 *	471.82 ± 79.74	422.12 ± 71.52 *	408.02 ± 65.06	60.06 ± 32.68	59.83 ± 28.72

* *p* ≤ 0.05.^#^ *p* ≤ 0.05. On the basis sex.

**Table 7 ijerph-20-01263-t007:** Results of daily physical activity levels and sedentary behavior variables on the basis of centre ownership and sex as independent variables of analysis.

				Sedentary (min)		Light PA (min)		Moderate PA (min)		Vigorous PA (min)		MVPA (min)	
	Type School	Pu.	Pri.	Pu.	Pri.	Pu.	Pri.	Pu.	Pri.	Pu.	Pri.	Pu.	Pri.
Male	(*n* = 542)	271	271	561.17 ± 189.98	594.89 ± 191.99	205.95 ± 68.07	208.99 ± 64.84	74.57 ± 38.83	76.15 ± 38.09	10.12 ± 9.00 ^###^	9.80 ± 8.76 ^###^	84.70 ± 46.43	85.96 ± 45.41
Female	(*n* = 540)	275	265	563.06 ± 185.30	580.11 ± 178.99	216.29 ± 67.07 ^#^	215.29 ± 63.77	83.39 ± 35.50 ^##^	79.37 ± 35.14	7.19 ± 5.81	6.64 ± 5.84	90.58 ± 39.82 ^#^	86.01 ± 39.65

On the basis of centre ownership ^#^ *p* ≤ 0.05; ^##^ *p* ≤ 0.01; ^###^
*p* ≤ 0.001. On the basis sex.

## Data Availability

Data supporting reported results can be found by mailing authors.

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
