# Peer review of "Physical Activity Levels and Sleep in Schoolchildren (6–17) with and without School Sport"

_ijerph, 2023, doi:10.3390/ijerph20021263_

Round 1
Reviewer 1 Report
This paper describes physical activity and sleep in youth from the Basque Country (n=1082). The main objective was to compare these behaviors between those participating in sports in the school environment to those who do not. Although the data collected are important, some points need to be addressed. Moreover, I have some suggestions that will need extensive changes.
1) The rationale for the study requires further examination of the bidirectional relationship between physical activity and sleep based on relevant literature
2) How does the transition from childhood to adolescence affect both physical activity and sleep? This need to be added in the introduction section.
3) Participation in sports activity in the school environment need to be detailed in the methods section (type of sport, hours per week, time of practice…) Is there any evolution in term of practice according to age?
4) The authors should outline the difference between youth practicing school sports with those who don’t for both physical activities and sleep according to age (6-12 vs 12-17 yrs. old). This is an important result that needs to add to the main paper instead of the supplementary file. How do you interpret and explain these results?
5) The discussion section is brief and does not outline the difference between school sports practice and physical activity. To my opinion this part will need extensive changes :
- Please discuss the difference between school sports practice and physical activity. Is there any differential effect on sleep according to the literature and your results?
- Please discuss the results moved from the supplementary file.
- Please added a limitation section
Author Response
We would like to thank the reviewer for the comments and the time spent reviewing our manuscript. We attach in pdf format all the answers to the proposed comments. Best regards.

Reviewer 2 Report
Dear Authors,
Please find attached the document drafted from the revision I have made.
Kind regards

Author Response

(The authors gave the same response as above.)

Round 2
Reviewer 1 Report
no further comments.
Author Response
Dear reviewer. Thank you again for your time in reviewing the manuscript. With the suggestions made by you, we consider that the manuscript has gained in quality. All the best. Best regards.
Reviewer 2 Report
Dear Authors.
Thank you for taking into consideration the comments I made in the previous review. The improvements in the manuscript are substantial. In the attached document you can find the comments of the second revision.
Kind regards

Author Response
Dear reviewer. Thank you again for your time in reviewing the manuscript. With the suggestions made by you, we consider that the manuscript has gained in quality. Attached in pdf format are the responses to all the comments made in the second round of review. We hope they have met your expectations. All the best. Best regards.
